# CLASSIFIER GUIDANCE ENHANCES DIFFUSION-BASED ADVERSARIAL PURIFICATION BY PRESERVING PREDICTIVE INFORMATION

## ABSTRACT

Adversarial purification is one of the promising approaches to defend neural networks against adversarial attacks. Recently, methods utilizing diffusion probabilistic models have achieved great success for adversarial purification in image classification tasks. However, such methods fall into the dilemma of balancing the needs for noise removal and information preservation. This paper points out that existing adversarial purification methods based on diffusion models gradually lose sample information during the core denoising process, causing occasional label shift in subsequent classification tasks. As a remedy, we suggest to suppress such information loss by introducing guidance from the classifier confidence. Specifically, we propose Classifier-cOnfidence gUided Purification (COUP) algorithm, which purifies adversarial examples while keeping away from the classifier decision boundary. Experimental results show that COUP can achieve better adversarial robustness under strong attack methods.

## 1 INTRODUCTION

Extensive research has shown that neural networks are vulnerable to well-designed adversarial examples, which are created by adding imperceptible perturbations on benign samples (Goodfellow et al., 2014; Madry et al., 2017; Brendel et al., 2017; Croce & Hein, 2020). Various approaches have been explored to improve model robustness, including model training enhancement (Madry et al., 2017; Zhang et al., 2019; Gowal et al., 2021) and input data preprocessing (Samangouei et al., 2018; Li & Ji, 2019). While these works have significantly improved adversarial robustness, there is still a clear gap in the classification accuracy between clean and adversarial data.

In recent years, adversarial purification with diffusion probabilistic models (Nie et al., 2022; Xiao et al., 2022; Carlini et al., 2022; Wu et al., 2022; Wang et al., 2022) has become an effective approach to defend against adversarial attacks in image classification tasks. The key idea is to preprocess the input image using an auxiliary diffusion model before feeding it into the downstream classifier. Leveraging the strong ability of generative models to fit data distributions, adversarial purification methods are able to purify the adversarial examples by pushing them toward the manifold of benign data. Such a process is essentially a denoising process, which gradually removes possible noise from input data.

Though achieved advanced performance on robust image classification tasks, adversarial purification methods rely solely on the denoising function during purification, thus inevitably fall into the dilemma of balancing the need for noise removal and information preservation (Nie et al., 2022). While stronger purification may destroy the image details that are necessary for classification, weaker purification may not be sufficient to remove the adversarial perturbations completely. The passive strategy to balance this, i.e. controlling the global purification steps (Nie et al., 2022), is limited in its effect, in the sense that data information monotonically loses as the purification steps grow. The existing method to mitigate the loss of information is to constrain the distance between the input adversarial example and the purified image (Wang et al., 2022; Wu et al., 2022). However, such constraint may inhibit the purified example from escaping the adversarial region effectively.

In this paper, we aim to propose a method that directly takes into consideration the need for information preservation. We borrow the idea of classifier guidance for diffusion models (Song et al.,

2020b; Ho & Salimans, 2022; Dhariwal & Nichol, 2021; Kawar et al., 2022), using the classifier confidence on the current class label $y$ given data $x$ as an indicator of the degree of preservation and try to maintain high confidence during the purification process. Staying away from low-confidence areas is beneficial to successful purification since such areas are close to the decision boundary and are more sensitive to small perturbations. Approaching a low confidence area can result in a potential label shift problem, i.e. a sample that initially has the correct label is misclassified after purification, especially when combined with stochastic defense strategies.

Specifically, we propose a Classifier-cOnfidence gUided Purification algorithm (COUP) with diffusion models to match the requirement of information preservation. The key idea is to gradually push input data towards high-likelihood regions while keeping relatively high confidence for classification. This process is realized by applying the denoising process together with a regularization term which improves the confidence score of the downstream classifier. This guidance discourages the purification process from moving toward decision boundaries, where the classifier becomes confused, and the confidence decreases.

We empirically evaluate our algorithm on CIFAR-10 (Krizhevsky et al., 2009) dataset using strong adversarial attack methods, including AutoAttack (Croce & Hein, 2020), which contains both white-box and black-box attacks, Backward Pass Differentiable Approximation (BPDA) (Athalye et al., 2018), as well as EOT (Athalye et al., 2018) to tackle the randomness in defense strategy. Results show that COUP outperforms DiffPure (Nie et al., 2022), in terms of adversarial robustness under $l_2$ and $l_\infty$ constraints.

Our work has the following main contributions:

- We propose a new adversarial purification algorithm COUP. By leveraging the confidence score from the downstream classifier, COUP is able to preserve the predictive information while removing the malicious perturbation.

- We provide both theoretical and empirical analysis for the effect of confidence guidance, showing that keeping away from the decision boundary can preserve predictive information and alleviate the label shift problem which are beneficial for classification.

- Experiments demonstrate that COUP can achieve significantly higher adversarial robustness against strong attack methods, reaching a robustness score of $73.05\%$ for $l_\infty$ and $83.13\%$ for $l_2$ under AutoAttack method on CIFAR-10 dataset.

## 2 RELATED WORK

**Adversarial Training** Adversarial training consolidates the discriminative model by enriching trained data (Madry et al., 2017). Such methods include generating adversarial examples during model training (Madry et al., 2017; Zhang et al., 2019; Qin et al., 2019; Jiang et al., 2019), or using an auxiliary generation model for data augmentation (Gowal et al., 2021; Rebuffi et al., 2021; Wang et al., 2023). Though effective, such methods still face adversarial vulnerability for unseen threats (Laidlaw et al., 2020) and suffer from computational complexity (Wong et al., 2020) during training. These works are orthogonal to ours and can be combined with our purification method.

**Adversarial Purification** Adversarial purification is another effective approach to defense. The idea is to use a generative model to purify the adversarial examples before feeding them into the discriminative model for classification. Based on different generative models (Goodfellow et al., 2020; Kingma & Welling, 2013; LeCun et al., 2006; Ho et al., 2020; Song et al., 2020b), corresponding purification methods are proposed (Samangouei et al., 2018; Li & Ji, 2019; Du & Mordatch, 2019; Grathwohl et al., 2019; Hill et al., 2020; Carlini et al., 2022) to convert the perturbed sample into a benign one. Recently, adversarial purification methods based on diffusion models have been proposed and achieved better performance (Yoon et al., 2021; Xiao et al., 2022; Nie et al., 2022; Wu et al., 2022; Wang et al., 2022). Among these works, DiffPure (Nie et al., 2022) achieves the most remarkable result, which is the focus of our comparison.

**Classifier Guided Diffusion Models** Diffusion models (Sohl-Dickstein et al., 2015; Ho et al., 2020; Song & Ermon, 2019; Song et al., 2020b) are recently proposed generation models, achieving high generation quality on images. Some works further leverage the guidance of the classifier to achieve controllable generation and improve the image synthesis ability (Ho & Salimans, 2022; Song et al.,

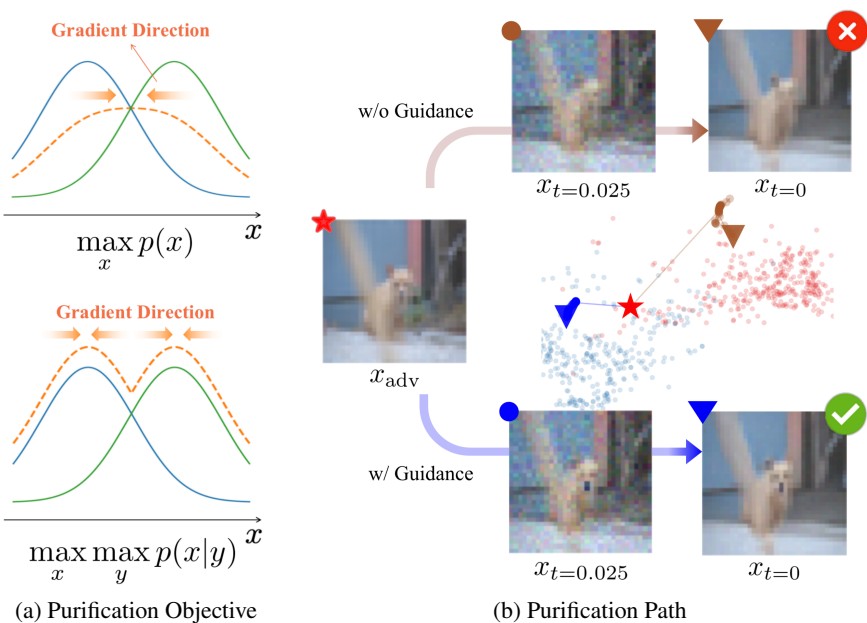

(a) Purification Objective        (b) Purification Path

Figure 1: The distinction between the existing purification method and classifier-confidence guided purification is elucidated in terms of (a) purification objective and (b) visualization of the purification path and the resultant purified image. In (a), the blue curve and green curve represent $p(x|y = 0)$ and $p(x|y = 1)$, respectively. The orange dotted line indicates the optimization objective ($p(x)$ or $\max_y p(x|y)$) and the direction of the gradient is shown accordingly by the orange arrow. This comparison reveals that guiding by classifier confidence is crucial to purify the example toward the category center. The purified image in (b) suggests that classifier guidance is advantageous for preserving predictive information pivotal for robust classification.

2020b; Dhariwal & Nichol, 2021; Kawar et al., 2022). Although the idea of classifier guidance has been proven to be beneficial for better image generation quality, whether the guided diffusion is helpful for adversarial purification is not yet verified. In our work, we utilize the classifier guidance to mitigate the loss of predictive information, so as to strike a balance between information preservation and purification.

## 3   Objective of Adversarial Purification

In this section, we present an objective of adversarial purification from the perspective of classification tasks and discuss how to achieve such an objective. The analysis results indicate the importance of considering the need for information preservation directly, which can be achieved by introducing guidance from classifier confidence during the purification process.

The concept of adversarial examples is first proposed by Szegedy et al. (2013), showing that neural networks are vulnerable to imperceptible perturbations. Data $x_{\mathrm{adv}}$ is called an adversarial example w.r.t. $x \in \mathbb{R}^d$, if it is close enough and belongs to the same class $y_{\mathrm{true}}$ under ground truth classifier $p(y|\cdot)$, but has a different label under model $\hat{p}(y|\cdot)$, such that

$$\arg\max_y \hat{p}(y|x_{\mathrm{adv}}) \neq \arg\max_y \hat{p}(y|x) = y_{\mathrm{true}}, \ \|x_{\mathrm{adv}} - x\| \leq \epsilon. \tag{1}$$

The idea of adversarial purification is to introduce a purification process before feeding data into the classifier. Though the optimal purification result would be converting $x_{\mathrm{adv}}$ back to $x$, it is almost impossible and not necessary. For the task of classification, it is sufficient as long as $x_{\mathrm{adv}}$ shares the same label with $x$. Therefore, the objective of adversarial purification from the classification perspective can be formulated as

$$\max_r \ \hat{p}(y_{\mathrm{true}}|r(x_{\mathrm{adv}})), \tag{2}$$

where $r(\cdot) : \mathbb{R}^d \to \mathbb{R}^d$ is the purification function.

In practice, the above objective cannot be optimized directly since the ground truth label $y_{\text{true}}$ is in general unknown, and so is the clean data $\boldsymbol{x}$. A substitute idea may be using the label of a nearby but not too close data $\boldsymbol{x}'$ with high probability density $p(\boldsymbol{x}')$ instead. The reasons include two aspects: first, the classification model is more trustworthy in high-density areas, thus can eliminate the effect of adversarial perturbation; second, compared with far-away data, a data with moderate distance from $\boldsymbol{x}_{\text{adv}}$ is more likely to share the same label with the clean data $\boldsymbol{x}$. The search for such nearby data is non-trivial, therefore as an alternative, we can use $r(\boldsymbol{x}_{\text{adv}})$ itself as the nearby data and take $\arg\max_y \ \hat{p}(y|r(\boldsymbol{x}_{\text{adv}}))$ as an approximation of $y_{\text{true}}$.

According to this idea, the ideal $r(\boldsymbol{x}_{\text{adv}})$ for solving Eq.(2) should balance the following requirements:

1. Maximizing the probability density $p(r(\boldsymbol{x}_{\text{adv}}))$. This helps to remove the adversarial noise.

2. Maximizing the classifier confidence $\max_y \ \hat{p}(y|r(\boldsymbol{x}_{\text{adv}}))$. This helps to preserve the essential information for classification.

3. Controlling the distance $\|r(\boldsymbol{x}_{\text{adv}}) - \boldsymbol{x}_{\text{adv}}\|$. This helps to avoid significant semantic changes.

**Discussion of existing works.** Existing adversarial purification methods usually utilize a generative model $\hat{p}(\boldsymbol{x})$ to approximate $p(\boldsymbol{x})$, and try to maximize $\hat{p}(r(\boldsymbol{x}_{\text{adv}}))$ while keeping the distance $\|r(\boldsymbol{x}_{\text{adv}}) - \boldsymbol{x}_{\text{adv}}\|$ small enough. We show a few examples here. DefenseGAN (Samangouei et al., 2018) is an early work for such purification, it uses generative adversarial nets as a generative model and optimizes $\min_{\mathbf{z}} \|G(\mathbf{z}) - \boldsymbol{x}_{\text{adv}}\|_2$ for purification. The $l_2$ norm is used for controlling the distance, and $r(\boldsymbol{x}_{\text{adv}}) = G(\mathbf{z})$ is guaranteed to have a high probability with the generator $G(\cdot)$. DiffPure (Nie et al., 2022) is a recently proposed adversarial purification method, it uses diffusion probabilistic models as a generative model. The purification process is a stochastic differential equation, whose main part includes a score function update which essentially increases the probability density of $r(\boldsymbol{x}_{\text{adv}})$. Meanwhile, it has been shown that the $l_2$ distance is implicitly controlled by the global update steps.

We find that the requirement of classifier confidence maximization is widely overlooked in existing works. A possible explanation might be that this requirement is partially addressed through density maximization. This is the case when there is little overlap between different classes of data, such that high probability density generally means high classifier confidence. However, when such overlap exists, i.e. the decision boundary crosses high probability density areas, maximizing probability density alone can be problematic. Consider the case where areas around the decision boundary have the highest density, the purification process without classifier confidence guidance will drive nearby samples towards the boundary, causing potential label shift especially when combined with stochastic defense strategies. A specific example is shown in Fig. 1. As a result, we suggest directly addressing the need for information preservation by maximizing the classifier confidence simultaneously.

## 4    CLASSIFIER-CONFIDENCE GUIDED PURIFICATION

Motivated by the objective of adversarial purification discussed in Section 3, we propose a **C**lassifier-c**O**nfidence g**U**ided **P**urification (COUP) method with score-based diffusion models to achieve the objective of adversarial purification.

### 4.1    METHODOLOGY

In order to meet the three requirements in Section 3, we address each of them separately: we use a score-based diffusion model and apply the denoising process to maximize the likelihood of purified image (i.e. $\max_{\boldsymbol{r}} \hat{p}(\boldsymbol{r}(\boldsymbol{x}_{\text{adv}}))$); we query the classifier during purification and maximize the classifier confidence (i.e. $\max_{\boldsymbol{r}} \max_y \hat{p}(y|\boldsymbol{r}(\boldsymbol{x}_{\text{adv}}))$); we control the distance by choosing appropriate global update steps $t^*$. We will first introduce the diffusion model used for denoising, and then explain how we utilize the classifier confidence for adversarial purification.

**Score-based Diffusion Models** Diffusion probabilistic models are deep generative models that have recently shown remarkable generation ability. Among existing diffusion models, Score SDE is a unified architecture that models the diffusion process by a stochastic differential equation (SDE)

and the denoising process by a corresponding reverse-time SDE. Specifically, the forward SDE is formalized as

$$\mathrm{d}\boldsymbol{x} = \boldsymbol{f}(\boldsymbol{x}, t)\mathrm{d}t + g(t)\mathrm{d}\boldsymbol{w}. \tag{3}$$

where $t$ is the time, $\boldsymbol{f}(\boldsymbol{x}, t)$ is the drift function, $g(t)$ is the diffusion coefficient, $\boldsymbol{w}$ is a standard Wiener process (Brownian motion). The effect of forward SDE is to progressively inject Gaussian noise into the input, and eventually transfer the original data to a Gaussian distribution. The corresponding reverse-time SDE is

$$\mathrm{d}\boldsymbol{x} = [\boldsymbol{f}(\boldsymbol{x}, t) - g(t)^2 \nabla_{\boldsymbol{x}} \log p_t(\boldsymbol{x})]\mathrm{d}t + g(t)\mathrm{d}\overline{\boldsymbol{w}}, \tag{4}$$

where $\overline{\boldsymbol{w}}$ is a standard reverse-time Wiener process. $\nabla_{\boldsymbol{x}} \log p_t(\boldsymbol{x})$ is parameterized by a neural model $\boldsymbol{s}_{\boldsymbol{\theta}}(\boldsymbol{x}_t, t)$, which is also called the score function. The score function is the core of the reverse process, driving data $\boldsymbol{x}$ towards higher probability by improving $\log p_t(\boldsymbol{x})$, where $p_t(\boldsymbol{x})$ can be viewed as an approximation of $p(\boldsymbol{x})$.

**Adversarial Purification with Guidance of Classifier Confidence**

The reverse-time SDE has been used for adversarial purification in previous diffusion-based purification methods. Our key idea is to introduce the guidance signal $\max_y \hat{p}(y|\boldsymbol{x})$ into the reverse-time SDE, such that we use $\log \hat{p}(\boldsymbol{x}) + \lambda \cdot \log \max_y \hat{p}(y|\boldsymbol{x})$ to replace $\log \hat{p}(\boldsymbol{x})$ in the score function $\boldsymbol{s}_{\theta}(\boldsymbol{x}, t) = \nabla_{\boldsymbol{x}} \log \hat{p}(\boldsymbol{x})$. Therefore, the purification update rule becomes

$$\mathrm{d}\boldsymbol{x} = \{\boldsymbol{f}(\boldsymbol{x}, t) - g(t)^2 [\boldsymbol{s}_{\theta}(\boldsymbol{x}, t) + \lambda \underbrace{\nabla_{\boldsymbol{x}} \log \max_y \hat{p}(y|\boldsymbol{x})}_{\text{classifier guidance}}]\}\mathrm{d}t + g(t)\mathrm{d}\overline{\boldsymbol{w}}, \tag{5}$$

where $\hat{p}(y|\boldsymbol{x})$ is the classifier confidence estimated by a fully trained classifier. The coefficient $\lambda > 0$ is determined by how much we can trust the classifier. The more accurate the classifier is, the larger value we can take for $\lambda$. More discussions on the choice of $\lambda$ can be found in section E.1.

In practice, we use VP-SDE (Song et al., 2020b), such that the drift function and diffusion coefficient are

$$\boldsymbol{f}(\boldsymbol{x}, t) = -\frac{1}{2}\beta(t)\boldsymbol{x}, \quad g(t) = \sqrt{\beta(t)}, \tag{6}$$

where $\beta(t)$ is a linear interpolation from $\beta_{\min}$ to $\beta_{\max}$. The purification process starts from $\boldsymbol{x}_{t^*} = \boldsymbol{x}_{\mathrm{adv}}$ at time $t = t^*$ and ends at time $t = 0$ to get $\boldsymbol{x}_0$. The global purification steps $t^*$ controls the distance between $\boldsymbol{x}_{t^*}$ and $\boldsymbol{x}_0$, which we will later explain in section 4.3.

## 4.2 THE COUP ALGORITHM

According to the purification rule of Eq. 5, we design our Classifier-cOnfidence gUided Purification (COUP) algorithm in Algo. 1. Our algorithm first set up the drift function and diffusion scale of guided reverse-time SDE according to Eq. 6. Then, we adopt an SDE process from $t = t^*$ to $t = 0$ to get the purified image $\hat{\boldsymbol{x}}_{\mathrm{ben}}$, where the input is adversarial example $\boldsymbol{x}_{\mathrm{adv}}$. Finally, we can use the trained classifier to predict the label of the purified image $\hat{\boldsymbol{x}}_{\mathrm{ben}}$. We omit the forward diffusion process since it yields no positive impact on the objectives of adversarial purification discussed in Section 3, and may cause a potential semantic shift. A detailed discussion can be found in Appendix C.1.

Note that COUP can use the off-the-shelf diffusion model and the fully trained classifier. In other words, we combine the off-the-shelf generative model and the trained classifier to achieve higher robustness.

**Adaptive Attack** In order to evaluate our defense method against strong attacks, we follow Nie et al. (2022) to use an augmented SDE to compute the gradient of $\hat{\boldsymbol{x}}_{\mathrm{ben}}$ w.r.t. $\hat{\boldsymbol{x}}_{\mathrm{adv}}$ in COUP for gradient-based attacks, detailed in Appendix C.3. In other words, we expose our purification strategy to the attacker to obtain strict robustness evaluation. In this way, we can make a fair comparison with other adversarial defense methods. Specifically, we adapt our whole process including purification and classification to adaptive attack (Athalye et al., 2018; Tramer et al., 2020) according to the chain rule. Since $\hat{\boldsymbol{x}}_{\mathrm{ben}}$ is the input of classifier, $\frac{\partial \mathcal{L}}{\partial \hat{\boldsymbol{x}}_{\mathrm{ben}}}$ can be obtained easily obtained. Then we can get the full gradient $\frac{\partial \mathcal{L}}{\partial \boldsymbol{x}_{\mathrm{adv}}} = \frac{\partial \mathcal{L}}{\partial \hat{\boldsymbol{x}}_{\mathrm{ben}}} \cdot \frac{\partial \hat{\boldsymbol{x}}_{\mathrm{ben}}}{\partial \boldsymbol{x}_{\mathrm{adv}}}$.

---

**Algorithm 1** Classifier-Confidence Guided Purification (COUP) Algorithm.

---

**Input:** Perturbed example $\boldsymbol{x}_{\mathrm{adv}}$.
**Output:** Purified example $\boldsymbol{x}_{\mathrm{ben}}$, predicted label $\hat{y}$.
1 **Required:**Trained classifier $f_{\mathrm{cls}}(\boldsymbol{x}) \approx \max_y \hat{p}(y|\boldsymbol{x})$ score function $\boldsymbol{s}_{\boldsymbol{\theta}}(\boldsymbol{x}, t) \approx \nabla_{\boldsymbol{x}} \log p_t(\boldsymbol{x})$ of fully trained diffusion model, optimal timestep $t^*$, and a regularization parameter $\lambda$.
2   Set up drift function: $\boldsymbol{f}(\boldsymbol{x}, t) \leftarrow -\frac{1}{2}\beta(t)\boldsymbol{x} - \beta(t)\{\boldsymbol{s}_{\theta}(\boldsymbol{x}, t) + \lambda\nabla_{\boldsymbol{x}} \log[f_{\mathrm{cls}}(\boldsymbol{x})]\}$
3   Set up diffusion function: $g(t) \leftarrow \sqrt{\beta(t)}$
4   Solve SDE for Purification: $\hat{\boldsymbol{x}}_{\mathrm{ben}} \leftarrow \mathrm{SDE}(\boldsymbol{x}_{\mathrm{adv}}, \boldsymbol{f}(\boldsymbol{x}, t), g(t), t^*, 0)$
5   Classification: $\hat{y} \leftarrow \arg\max_y f_{\mathrm{cls}}(\hat{\boldsymbol{x}}_{\mathrm{ben}}, y)$
6   return $\hat{y}$

---

## 4.3 ANALYSIS OF COUP

In this section, we further analyze the effectiveness of our method. First, we show that under the guidance of classifier confidence, our method can better preserve information for classification. Second, under the guided reverse-time VP-SDE, the distance $\|r(\boldsymbol{x}_{\mathrm{adv}}) - \boldsymbol{x}_{\mathrm{adv}}\|$ can be bounded through controlling $t^*$.

To show that our proposed confidence guidance helps to preserve data information, we give theoretical analysis on a simple case where such guidance can be proved to alleviate the label shift problem. Consider a 1-dimension SDE $\mathrm{d}x = f(x,t)\mathrm{d}t + g(t)\mathrm{d}w$ with starting point $x_{t=0} = x_0 > 0$ and final solution $x_{t=1}$, which is simulated using the Euler method with step size $\Delta t = 1/n$. Denote as $P_{<0}(x_0, f, g)$ the label flip probability such that there exist $t^* \in [0, 1]$ satisfying $x_{t^*} < 0$, we have the following proposition:

**Proposition 4.1** *If for any $t \in [0, 1]$ and $x > 0$, there is $f_0(x, t) < f_1(x, t)$ and $f_0(x, t)$ is strictly monotonically increasing w.r.t. $x$, then*

$$P_{<0}(x_0, f_1, g) < P_{<0}(x_0, f_0, g). \tag{7}$$

Proposition 4.1 supports the claim that forces pushing the data away from the decision boundary are helpful to avoid the label shift problem. Consider the case where the data is composed of two classes: one distribution follows $N(\mu, \sigma^2)$ and another follows $N(-\mu, \sigma^2)$, the conditions in Proposition 4.1 would be satisfied using a VP-SDE (as $f_0$) and a corresponding SDE with guidance (as $f_1$), since the added gradient of classifier confidence is always positive on $(0, +\infty)$. In this case, Proposition 4.1 shows that with the guidance from the ground-truth classifier, it is less likely for a sample to change its label during the purification process. We provide the proof of proposition 4.1 in Appendix A.1.

Next, we show that the distance between the input sample $\boldsymbol{x}$ and the purified sample $r(\boldsymbol{x})$ can be bounded under our proposed method, thus can avoid severe semantic changes during purification. The result of proposition 4.2 indicates that for an an adversarial example $\boldsymbol{x}_{\mathrm{adv}}$, the distance $\|r(\boldsymbol{x}_{\mathrm{adv}}) - \boldsymbol{x}_{\mathrm{adv}}\|$ has an upper bound, which is monotonically increasing w.r.t. $t^*$. As a result, the maximal distance can be controlled by adjusting $t^*$.

**Proposition 4.2** *Under the assumption that $\|\boldsymbol{s}_{\boldsymbol{\theta}}(\boldsymbol{x}, t)\| \leq \frac{1}{2}C_s$, $\|\nabla_{\boldsymbol{x}} p(\cdot|\boldsymbol{x})\| \leq \frac{1}{2}C_p$, and $\|\boldsymbol{x}\| \leq C_x$, the denoising error of our guided reverse variance preserving SDE (VP-SDE) can be bounded as*

$$\|\boldsymbol{r}(\boldsymbol{x}) - \boldsymbol{x}\| \leq \sqrt{e^{2\gamma(t^*)} - 1}\,\|\boldsymbol{\epsilon}\| + \gamma(t^*)(C_s + C_p) + (e^{\gamma(t^*)} - 1)C_x, \tag{8}$$

$\gamma(t^*) := \int_0^{t^*} \frac{1}{2}\beta(s)\mathrm{d}s$, $\boldsymbol{\epsilon} \sim \mathcal{N}(\mathbf{0}, \mathbf{1})$ *is the noise added by the reverse-time Wiener process.*

## 5 EXPERIMENTS

In this section, we mainly evaluate the adversarial robustness of COUP against AutoAttack (Croce & Hein, 2020), including both black-box and white-box attacks. Furthermore, we analyze the mechanism of the classifier confidence guidance through a case study and ablation study to verify the effectiveness of COUP. Besides, we combine our work with the state-of-the-art adversarial training method for further promotion.

Table 1: Accuracy and Robustness against AutoAttack under $l_\infty(\epsilon = 8/255)$ and $l_2(\epsilon = 0.5)$ threat model on CIFAR-10. The model architecture is reverse-time VP-SDE with $t^* = 0.1$ for $l_\infty$ and $t^* = 0.075$ for $l_2$. We use the fully trained WideResNet-28-10 for the classifier.

<table>
<tr><td colspan="3" align="center">(a) AutoAttack under $l_\infty$</td><td colspan="3" align="center">(b) AutoAttack under $l_2$</td></tr>
<tr><td>Defense</td><td>Acc (%)</td><td>**Rob** (%)</td><td>Defense</td><td>Acc (%)</td><td>**Rob** (%)</td></tr>
<tr><td>-</td><td>96.09</td><td>0.00</td><td>-</td><td>96.09</td><td>0.00</td></tr>
<tr><td>Wu et al. (2020)</td><td>85.36</td><td>59.18</td><td>Rony et al. (2019)</td><td>89.05</td><td>66.41</td></tr>
<tr><td>Zhang et al. (2020)</td><td>89.36</td><td>59.96</td><td>Ding et al. (2018)</td><td>88.02</td><td>67.77</td></tr>
<tr><td>Rebuffi et al. (2021)</td><td>87.33</td><td>61.72</td><td>Wu et al. (2020)</td><td>88.51</td><td>72.85</td></tr>
<tr><td>Wu et al. (2020)</td><td>88.25</td><td>62.11</td><td>Sehwag et al. (2021)</td><td>90.31</td><td>75.39</td></tr>
<tr><td>Gowal et al. (2020)</td><td>89.48</td><td>62.70</td><td>Augustin et al. (2020)</td><td>92.23</td><td>77.93</td></tr>
<tr><td>Gowal et al. (2021)</td><td>87.50</td><td>65.24</td><td>Rebuffi et al. (2021)</td><td>91.79</td><td>78.32</td></tr>
<tr><td>Nie et al. (2022)</td><td>89.02</td><td>70.64</td><td>Nie et al. (2022)</td><td>91.03</td><td>78.58</td></tr>
<tr><td>Our COUP</td><td>**90.04**</td><td>**73.05**</td><td>Our COUP</td><td>**92.58**</td><td>**83.13**</td></tr>
</table>

## 5.1 EXPERIMENTAL SETTINGS

**Dataset and Models** We follow the setting of robustbench (Croce et al., 2020), an universal robustness evaluation benchmark. Since the robustness evaluation suffers from a long inference time, especially under the attacker's query behavior, we evaluate our method on CIFAR-10 (Krizhevsky et al., 2009) dataset. To make a fair comparison with other diffusion-based purification methods, we follow the settings of DiffPure (Nie et al., 2022), evaluating the robustness on 512 images under the same random seed. As for the purification model, we use variance preserving SDE (VP-SDE) of Score SDE (Song et al., 2020b) with $t^* = 0.1$ for $l_\infty$ threat model and $t^* = 0.075$ for $l_2$. We select two backbones of the classifier, including WRN-28-10 (WideResNet-28-10) and WRN-70-16 (WideResNet-70-16).

**Baselines** We compare our method with (1) robust optimization methods, that is the adversarial defense based on discriminative models, also including using generative models for data augmentation; (2) adversarial purification methods based on generative models before classification. Since some diffusion-based adversarial purification methods do not support gradient computation and do not design an adaptive attack, we compare the SOTA method (Nie et al., 2022) supported by AutoAttack (Croce & Hein, 2020).

**Evaluation Method** We evaluate our method against the AutoAttack (Croce & Hein, 2020) against the $l_\infty$ and $l_2$ threat models and Backward Pass Differentiable Approximation (BPDA) (Athalye et al., 2018). Since our method contains Brownian motion, we use both *standard* (including three white-box attacks APGD-ce, APGD-t, FAB-t, and one black-box attack Square) and *rand* mode (including two white-box attack APGD-ce, APGD-dlr with EOT=20) of AutoAttack, choosing the worse one to eliminate the 'fake robustness' brought by randomness. Since the white-box plays stronger attack behavior, we evaluate our algorithm across different classifier backbones and other analysis experiments against APGD-ce, one of the white-box in AutoAttack.

## 5.2 COMPARISON WITH RELATED WORK

**AutoAttack** We evaluate our COUP against AutoAttack and compare the robustness with other advanced defense methods according to the results proposed in robustbench (Croce et al., 2020). DiffPure (Nie et al., 2022) is the most related work to ours. According to the results in Table 1, our COUP achieves better robustness (+2.41% for $l_\infty$ and +4.55% for $l_2$ ) as well as better accuracy (+1.02% for $l_\infty$ and +1.55% for $l_2$), showing the effectiveness of classifier guidance enhancing the adversarial robustness through better purification. Owing to the adaptive attack, we can make a fair comparison with other purification algorithms as well as robust optimization methods based on discriminative models. The state-of-the-art method of robust optimization is an improved adversarial training using generated data by diffusion models. COUP is orthogonal to those. We will further discuss the comparison and combination with the SOTA work among them in Section 5.3.

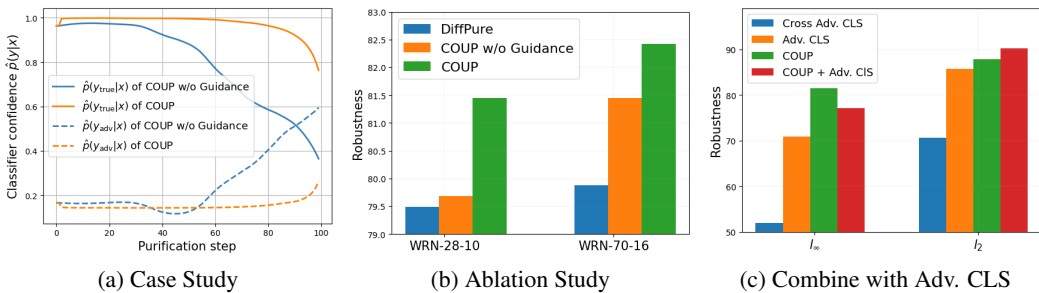

| (a) Case Study | (b) Ablation Study | (c) Combine with Adv. CLS |

Figure 2: (a) Expectation of disminative confidence $\hat{p}(y|\boldsymbol{x})$ of WRN-28-10 on label $y_{\text{true}}$ and adversarial label $y_{\text{adv}}$ over 14 bad cases of reverse-time SDE. (b) Robustness of different denoising methods, including the forward-reverse-time SDE (DiffPure) and the reverse-time SDE only (COUP and COUP w/o Guidance) under different classifier backbones. (c) Robustness of SOTA adversarial training method (Wang et al., 2023) (marked as Adv. CLS), in both the cross (trained under different $l_p$ norm with evaluation) and non-cross settings, and combined with our COUP against APGD-ce.

**BPDA** We also evaluate the robustness against BPDA (Athalye et al., 2018) in order to make a comparison with other guided diffusion-based purification methods (Wu et al., 2022; Wang et al., 2022) (since they do not support adaptive attack). Considering both Wu et al. (2022) and Wang et al. (2022) utilize adversarial samples as guidance, we compare with the more proficient GDMP (Wang et al., 2022) of the two (according to the results from Table 1 of Wu et al. (2022)). We test our method on CIFAR-10 dataset against PGD-20, employing a setting of $\epsilon = 8/255, \alpha = 0.007$. The experimental robustness of GDMP achieves 76.22%, DiffPure achieves 81.40%, and our COUP achieves the best of 83.20%. These results verify the effectiveness of our method against BPDA except for adaptive attack and obtain better performance than the adversarial examples guided diffusion-based purification algorithm.

### 5.3 EXPERIMENTAL ANALYSIS

**Case Study** In this part, we plot the curve of $\hat{p}(y_{\text{true}}|\boldsymbol{x})$ and $\hat{p}(y_{\text{adv}}|\boldsymbol{x})$ to show what happens from the view of classifier during purification, where the $\hat{p}(y|\boldsymbol{x})$ is the predict confidence by the classifier of label $y$. Moreover, we analyze the mechanism of the classifier guidance. To obtain the adversarial examples, we use APGD-ce under the $l_\infty$ threat model to attack the reverse-time SDE (COUP without Guidance) to get bad cases for COUP w/o Guidance. To focus on the purification process, we do not consider Brownian motion at inference time.

According to the analysis in Section 3, we use the predict confidence for ground truth label to evaluate the information preservation degree of the image during purification. Then we plot the curve as shown in Fig. 2(a). The rise of $\hat{p}(y_{\text{adv}}|\boldsymbol{x})$ is the reason for successful attack. Meanwhile, the decrease of $\hat{p}(y_{\text{true}}|\boldsymbol{x})$ shows that, during purification, predictive information keeps losing. After 90 steps of purification, $\hat{p}(y_{\text{true}}|\boldsymbol{x})$ suddenly declines to a very low level due to "over purification" and $\hat{p}(y_{\text{adv}}|\boldsymbol{x})$ dominates the prediction confidence, which leads to vulnerability. Next, we further explore the mechanism of classifier guidance. After adding classifier guidance, $\hat{p}(y_{\text{true}}|\boldsymbol{x})$ obtains a rapid rise under the guidance of the classifier at the very beginning. Besides, the guidance of the classifier alleviates the information loss (also weakens the influence of adversarial perturbation) during purification and finally results in correct classification.

**Analysis of Information Preservation on Toy Data** To verify the effectiveness of our method for information preservation, we use 2-Gaussian toy data and run a simulation of pure SDE and COUP to show that COUP can alleviate the label shift problem. The data distribution is a 1-dimension uniform mixture of 2 symmetric Gaussian distributions $\mathcal{N}(-0.5, 1)$ and $\mathcal{N}(0.5, 1)$, the data from which we label as $y = 0$ and $y = 1$, respectively. Starting from the point $x_0 = 0.2$, we apply the COUP algorithm with guidance weight ranging from 0 to 10.0. To simulate the adversarial vulnerability of the classifier, we use a noisy classifier $p(y = 1|x) = \frac{p_1(x)}{p_0(x) + p_1(x) + c \cdot n(x)}$, where $p_i(x)$ is the density function of class $i$, $n(x) = \frac{sin(100x)}{100}$ is the noise and $c$ is the noise level. We apply the Euler method

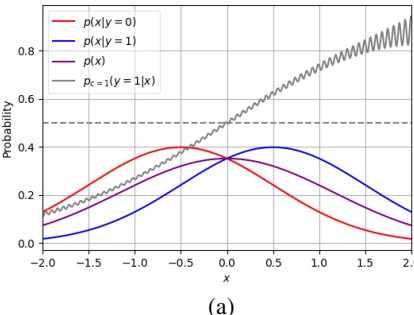 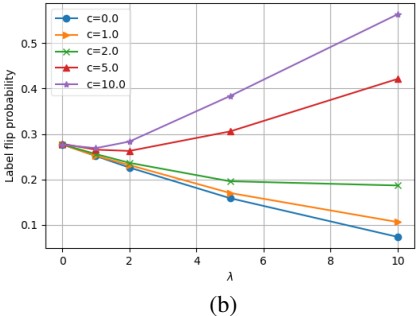

(a)                                              (b)

Figure 3: (a) The 2-Gaussian toy data and classifier with noise level $c = 1$. (b) Label flip probability under different noise levels $c$ and guidance weight $\lambda$ on 2-Gaussian toy data.

for SDE simulation, using step size $1e - 3$ and $t^* = 0.1$. We run $100,000$ times and estimate the label flip probability, i.e. $p(x_{t^*} < 0)$. The result in Fig. 3(b) shows that the guidance signal is overall helpful to keep the label unchanged under small or no noise. When the noise level is high, the classifier becomes untrustworthy. Thus, an appropriate $\lambda$ should be chosen.

**Analysis on Different Classifier Architectures** In order to evaluate the effectiveness of our guidance method for different architectures of classifiers, we adapt our purification method to both WRN-28-10 and WRN-70-16 against APGD-ce attack (under $l_\infty$). The results in Fig. 2(b) show that COUP achieves better robustness on both two classifier backbones. In other words, our method is effective across different classifier architectures.

**Ablation Study** In order to demonstrate the effectiveness of classifier guidance, we evaluate the robustness of COUP and COUP w/o Guidance (i.e. reverse-time SDE) against APGD-ce attack (under $l_\infty$) as an ablation. Results in Fig. 2(b) support that the robustness promotion in Table 1 of our COUP is mainly caused by the classifier guidance instead of the structure of diffusion model (since we remove the forward process from DiffPure).

**Comparison and Combination with SOTA of Adversarial Training** Considering the settings of robust evaluation, we argue that it is unfair to compare our COUP with the adversarial training algorithm. The reason is we do not make any assumption about the attack, while the adversarial training methods are specifically trained for the evaluation attack. Therefore, we additionally evaluate the SOTA (Wang et al., 2023) of adversarial training under both cross and non-cross settings. Specifically, in the cross-setting, we use the model trained for $l_2$ ($l_\infty$) to defend the attack under $l_\infty$ ($l_2$). The results in Table 2(c) show that it (Wang et al., 2023) suffers a severe robustness drop under the cross-setting. In other words, its robustness becomes poor when defending against unseen attacks.

Besides, to take advantage of their work (Wang et al., 2023), we combine our purification method with the adversarially trained classifier. When the classifier has better clean accuracy (95.16% under $l_2$), it can further improve the robustness against APGD-ce attack. However, worse accuracy under $l_\infty$ (92.44%) may provide inappropriate guidance for purification. Note that, in that case, our purification method COUP further improves their robustness from 70.90% to 77.15%.

## 6 CONCLUSION

Addressing the principal challenge in purification, i.e., achieving a balance between noise removal and information preservation, we employ the concept of the classifier-guided diffusion method. We discover that classifier-confidence guidance aids in preserving predictive information, which facilitates the purification of adversarial examples towards the category center, while the score function eliminates malicious perturbations. Specifically, we have introduced Classifier-cOnfidence gUided Purification (COUP) and have assessed its performance against AutoAttack and BPDA, comparing it with other advanced defense algorithms under the RobustBench benchmark. The results demonstrated that our COUP achieved superior adversarial robustness.

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

In Appendix A, we give the proofs of proposition 4.1 and proposition 4.2. In Appendix B, we provide supplementary analysis to show the necessity for introducing the guidance of classifier confidence, stated in the second requirement in Section 3. In Appendix C we further discuss the evidence of omitting the forward process and why the first requirement in Section 3 "Maximizing the probability density of purified image helps to remove the adversarial noise" is valid. Then we provide the details of adaptive attack for our method in Section 6. In Appendix D we show more details for implementation. In Appendix E we provide more experimental results including the effect of hyperparameters $t^*$ and $\lambda$, robustness against salt-and-pepper noise, and the speed test. Moreover, we further discuss the effectiveness of our method through the purified images and find that the semantic drift and information loss caused by DiffPure and COUP w/o Guidance, respectively. Finally, we analyze the limitations in Appendix F and broader impacts in Appendix G.

## A PROOFS OF PROPOSITIONS

### A.1 PROOF OF PROPOSITION 4.1

**Porposition 4.1** *(restated)* If for any $t \in [0, 1]$ and $x > 0$, there is $f_0(x, t) < f_1(x, t)$ and $f_0(x, t)$ is strictly monotonically increasing w.r.t. $x$, then

$$P_{<0}(x_0, f_1, g) < P_{<0}(x_0, f_0, g). \tag{9}$$

*Proof:* Define the noise term $g(t_i) z \sqrt{\Delta t}$ in update step $i$ as $z_i$, where $z \sim \mathcal{N}(0, 1)$ is a standard Gaussian noise so that each update by Euler method is

$$x_{t_{i+1}} = f(x_{t_i}, t_i) \Delta t + z_i.$$

Define an auxiliary function

$$\hat{f}(x, t) \triangleq \left\{ \begin{array}{ll} f(x, t) \Delta t, & if\ f(x, t) > 0, \\ -\infty, & otherwise, \end{array} \right.$$

then whether there exist $t^* \in [0, 1]$ such that $x_{t^*} < 0$ can be written as

$$c(x_0, f, z_0, \cdots, z_{n-1}) \triangleq \mathbb{I}(\hat{f}(\cdots \hat{f}(\hat{f}(x_0, 0) + z_0) + z_1 \cdots) + z_{n-1} < 0),$$

where $\mathbb{I}(\cdot)$ is the indicator function. Therefore, the label flip probability will be

$$P_{<0}(x_0, f, g) = \mathbb{E}_{z_0} \cdots \mathbb{E}_{z_{n-1}} c(x_0, f, z_0, \cdots, z_{n-1}).$$

Since there is $f_1(x, t) > f_0(x, t)$ for any $x > 0$, so that $\hat{f}_1(x_0, t) + z_0 \geq \hat{f}_0(x_0, t) + z_0$. The equivalence holds only when $f_1(x_0, t) \leq 0$. Considering $f$ is strictly monotonically increasing in $(0, +\infty)$, there is

$$\hat{f}_1(\hat{f}_1(x_0, t) + z_0) \geq \hat{f}_0(\hat{f}_1(x_0, t) + z_0)$$
$$\geq \hat{f}_0(\hat{f}_0(x_0, t) + z_0).$$

By parity of reasoning, we get

$$\hat{f}_1(\cdots \hat{f}_1(\hat{f}_1(x_0, 0) + z_0) + z_1 \cdots) + z_{n-1} \geq \hat{f}_0(\cdots \hat{f}_0(\hat{f}_0(x_0, 0) + z_0) + z_1 \cdots) + z_{n-1},$$

so that

$$c(x_0, f_1, z_0, \cdots, z_{n-1}) \leq c(x_0, f_0, z_0, \cdots, z_{n-1}),$$

thus

$$P_{<0}(x_0, f_1, g) \leq P_{<0}(x_0, f_0, g),$$

The above equivalence holds only when $P_{<0}(x_0, f_1, g) = 1$, which will not be possible due to the random nature of Gaussian noise, since there is always a positive probability such that $x_t > 0$ is kept during each update. So that

$$P_{<0}(x_0, f_1, g) < P_{<0}(x_0, f_0, g).$$

A.2 PROOF OF PROPOSITION 4.2 :

**Proposition 4.2** *(restated)* Under the assumption that $\|s_\theta(x,t)\| \le \frac{1}{2}C_s$, $\|\nabla_x p(\cdot|x)\| \le \frac{1}{2}C_p$, and $\|x\| \le C_x$, the denoising error of our guided reverse variance preserving SDE (VP-SDE) can be bounded as

$$\|r(x) - x\| \le \sqrt{e^{2\gamma(t^*)} - 1}\,\|\epsilon\| + \gamma(t^*)(C_s + C_p) + (e^{\gamma(t^*)} - 1)C_x, \tag{10}$$

$\gamma(t^*) := \int_0^{t^*} \frac{1}{2}\beta(s)\mathrm{d}s$, $\epsilon \sim \mathcal{N}(0,1)$ is the noise added by the reverse-time Wiener process.

*Proof:* Suppose given an example $x$, the COUP purification function $r(x)$ is a guided reverse-time SDE (depicted in Section 4.1) from $t = t^*$ to $t = 0$. Thus, the $l_2$ distance between $x$ and the purified example $r(x)$ is

$$
\begin{aligned}
&\|r(x) - x\| \\
&= \left\| x + \int_{t^*}^0 -\frac{1}{2}\beta(t)\left[ x_t + 2s_\theta(x_t, t) + 2\nabla_{x_t}\log[\max_y p(y|x_t)] \right]dt + \int_{t^*}^0 \sqrt{\beta(t)}d\overline{w} - x \right\| \\
&\le \left\| x + \int_{t^*}^0 -\frac{1}{2}\beta(t)x_t dt + \int_{t^*}^0 \sqrt{\beta(t)}d\overline{w} - x \right\| \\
&\quad + \left\| \int_{t^*}^0 -\beta(t)s_\theta(x_t, t)dt \right\| + \left\| \int_{t^*}^0 -\beta(t)\nabla_{x_t}\log[\max_y p(y|x_t)dt \right\|.
\end{aligned}
\tag{11}
$$

Under the assumption that $\|s_\theta(x,t)\| \le \frac{1}{2}C_s$ and $\|\nabla_x p(\cdot|x)\| \le \frac{1}{2}C_p$ , we have

$$\left\| \int_{t^*}^0 -\beta(t)s_\theta(x_t, t)dt \right\| + \left\| \int_{t^*}^0 -\beta(t)\nabla_{x_t}\log[\max_y p(y|x_t)dt \right\| \le \gamma(t^*)(C_s + C_p), \tag{12}$$

where $\gamma(t^*) := \int_0^{t^*} \frac{1}{2}\beta(s)ds$ and $\beta(t)$ is an linear interpolation with $\beta(0) = 0.1$, $\beta(1) = 20$. Since $\{x + \int_{t^*}^0 -\frac{1}{2}\beta(t)x_t dt + \int_{t^*}^0 \sqrt{\beta(t)}d\overline{w}\}$ is an integration of linear SDE, the solution $\hat{x}(0)$ follows a Gaussian distribution with $\mu$ and $\Sigma$ satisfies

$$
\begin{aligned}
\frac{d\mu}{dt} &= -\frac{1}{2}\beta(t)\mu \\
\frac{d\Sigma}{dt} &= -\beta(t)\Sigma + \beta(t)I_d.
\end{aligned}
\tag{13}
$$

The initial value is $\mu(t^*) = x(t^*)$ and $\Sigma(t^*) = 0$, thus, the result of solving Eq. 13 is $\mu(0) = e^{\gamma(t^*)}$ and $\Sigma(0) = e^{2\gamma(t^*)} - 1$. Therefore, the conditional probability $P(\hat{x}(0)|x(t^*) = x) \sim \mathcal{N}\left(e^{\gamma(t^*)}x, \left(e^{2\gamma(t^*)} - 1\right)I_d\right)$, the predicted $\hat{x}(0)$ of the linear SDE from $t = t^*$ to $t = 0$ can be represented as

$$\hat{x}(0) = e^{\gamma(t^*)}x + \sqrt{e^{2\gamma(t^*)} - 1}\epsilon, \tag{14}$$

where $\epsilon \sim \mathcal{N}(0,1)$.

Thus, we have

$$\|\hat{x}(0) - x\| = \left\| (e^{\gamma(t^*)} - 1)x + \sqrt{e^{2\gamma(t^*)} - 1}\epsilon \right\|. \tag{15}$$

Under the assumption that $\|x\| \le C_x$, we have

$$\|\hat{x}(0) - x\| \le \sqrt{e^{2\gamma(t^*)} - 1}\,\|\epsilon\| + (e^{\gamma(t^*)} - 1)C_x. \tag{16}$$

Therefore, we have

$$\|r(x) - x\| \le \sqrt{e^{2\gamma(t^*)} - 1}\,\|\epsilon\| + \gamma(t^*)(C_s + C_p) + (e^{\gamma(t^*)} - 1)C_x. \tag{17}$$

So far, the result in proposition 4.2 has been proved. It supports that under our purification method, the $l_2$ distance between the input and the purified sample can be upper bounded to avoid unexpected semantic changes.

# B ADDITIONAL ANALYSIS ON THE OBJECTIVE OF ADVERSARIAL PURIFICATION

In this section, We provide supplementary explanations for the content in Section 3.

## B.1 THE RELATION BETWEEN PROBABILITY DENSITY AND CLASSIFIER CREDIBILITY

We give an analysis of a simple case to show that the classification model is more trustworthy in high-density areas. Consider a binary classification problem with label space $y \in \{0, 1\}$ the ground truth classifier is $p(y|x)$ and the learned classification model is $\hat{p}(y|x)$. Assume the total numbers of training data is $n$ and all data are equally divided into each class. The value of $\hat{p}(y = 1|x)$ is determined by the proportion of sampled data in class $y = 1$ over total $2n$ samples in the neighborhood of $x$, i.e. $(x - \Delta x/2, x + \Delta x/2)$. Consider the points on the ground truth class boundary, i.e. $p(y|x) = \frac{1}{2}$, there is

$$p(x|y) = \frac{p(x) \cdot p(y|x)}{p(y)} = \frac{p(x) \cdot \frac{1}{2}}{\frac{1}{2}} = p(x). \tag{18}$$

According to the central limit theorem, as long as $n$ is large enough, the number of samples in class $y = 1$ would be following a Gaussian distribution with mean and variance as

$$\begin{aligned} \mu &= n \cdot p(y = 1|x)\Delta x, \\ \sigma^2 &= n \cdot p(y = 1|x)\Delta x \cdot (1 - p(y = 1|x)\Delta x). \end{aligned} \tag{19}$$

Applying Eq. 18, we get

$$\begin{aligned} \mu &= n \cdot p(x)\Delta x, \\ \sigma^2 &= n \cdot p(x)\Delta x \cdot (1 - p(x)\Delta x). \end{aligned} \tag{20}$$

Therefore the estimated model would be

$$\begin{aligned} \hat{p}(y|x) &\approx \frac{n \cdot p(x)\Delta x + \sqrt{n \cdot p(x)\Delta x \cdot (1 - p(x)\Delta x)}\epsilon_1}{2n \cdot p(x)\Delta x + \sqrt{n \cdot p(x)\Delta x \cdot (1 - p(x)\Delta x)}(\epsilon_1 + \epsilon_2)} \\ &= \frac{1 + \sqrt{\frac{1}{n}(\frac{1}{p(x)\Delta x} - 1)}\epsilon_1}{2 + \sqrt{\frac{1}{n}(\frac{1}{p(x)\Delta x} - 1)}(\epsilon_1 + \epsilon_2)}, \end{aligned} \tag{21}$$

where $\epsilon_1$ and $\epsilon_2$ are independent variables following a standard normal distribution. To estimate the approximate variance of $\hat{p}(y|x)$, we consider the value of $\epsilon_1$ and $\epsilon_2$ are limited within a range such that $\epsilon_1, \epsilon_2 \in (-c, c)$. Under such assumption, there is

$$\hat{p}(y|x) \in \frac{1}{2} \pm \frac{c}{2}\sqrt{\frac{1}{n}\left(\frac{1}{p(x)\Delta x} - 1\right)}. \tag{22}$$

So that the larger $p(x)$ is, the smaller variance of $\hat{p}(y|x)$ would be. As a result, it is necessary to seek high density areas where the classification boundary would be more stable.

## B.2 THE RELATION BETWEEN PROBABILITY DENSITY AND CLASSIFIER CONFIDENCE

We claimed in section 3 that optimizing the probability density is sometimes consistent with optimizing the classifier confidence. We will explain here when such relation exists and when it does not.

Still consider a binary classification task for 1-dimension data $x$ and uniformly distributed $y$, there is

$$p(y = 1|x) = \frac{2p(x|y = 1)}{p(x|y = 1) + p(x|y = 0)}. \tag{23}$$

Consider how $p(y = 1|x)$ changes as $p(x)$ changes:

$$\frac{\partial p(y = 1|x)}{\partial x} = 2 \cdot \frac{\frac{\partial p(x|y=1)}{\partial x}p(x|y = 0) - \frac{\partial p(x|y=0)}{\partial x}p(x|y = 1)}{[p(x|y = 1) + p(x|y = 0)]^2}. \tag{24}$$

Assuming

$$\frac{\partial p(x)}{\partial x} = \frac{1}{2} \left( \frac{\partial p(x|y=1)}{\partial x} + \frac{\partial p(x|y=0)}{\partial x} \right) > 0, \tag{25}$$

then whether $\frac{\partial p(y=1|x)}{\partial x}$ is also positive may not be sure. If data from different classes are well-separated, such that $\frac{\partial p(x|y=1)}{\partial x} > 0$ and $\frac{\partial p(x|y=0)}{\partial x} < 0$, there would be $\frac{\partial p(y=1|x)}{\partial x} > 0$. Otherwise, if $\frac{\partial p(x|y=1)}{\partial x} < 0$ and $\frac{\partial p(x|y=0)}{\partial x} > 0$, there would be $\frac{\partial p(y=1|x)}{\partial x} < 0$.

In conclusion, for most cases in a dataset with well-separated classes, optimizing $p(x)$ has similar effect as optimizing $p(y|x)$. However, there are exceptions that such objectives are opposite.

## C  ADDITIONAL ANALYSIS OF OUR METHODOLOGY

### C.1  EFFECT OF FORWARD PROCESS FOR ADVERSARIAL PURIFICATION

**Motivation of removing forward process** The objective of the forward process (injecting random noise) is to guide the training procedure for the reverse process. During adversarial purification, the model aims to recover the true label by maximizing likelihood, utilizing the score function of the backward process. Our method exclusively employs the reverse process, avoiding the forward process due to concerns over potential semantic changes induced by the random noise injection.

Existing adversarial purification methods (Nie et al., 2022; Xiao et al., 2022; Carlini et al., 2022; Wu et al., 2022; Wang et al., 2022; Yoon et al., 2021), follow the forward-reverse process of image generation tasks. However, they neither offer ablation studies nor provide theoretical evidence for the necessity of incorporating the forward process when the backward process is already in use. The habitual combination of forward and backward processes in applying diffusion models may not be a deliberate choice, warranting further consideration and evaluation.

Better performance (see Fig. 2(b)) after removing the forward process does not conflict with the existing works. To put it more rigorously, the conclusion drawn from existing works (Nie et al., 2022; Xiao et al., 2022; Carlini et al., 2022; Wu et al., 2022; Wang et al., 2022; Yoon et al., 2021) is that including random noise is necessary for successful purification. Note that both the forward and backward processes contain a noise-adding operation, retaining either process can introduce considerable random noise. Yoon et al. (2021) is an example, which applied the forward process but removed the noise term in the backward process. Our work applies the backward process with noise term by the Brownian motion, which is empirically sufficient to submerge the adversarial perturbation.

**Evidence** Therefore, we choose to discard the forward process since it could potentially lead to unexpected category drift, increasing the risk of misclassification. The case study in Fig. 5 in Appendix E.4 illustrates such a difference, where DiffPure causes more of a semantic change than COUP w/o guidance. For numerical evidence, we empirically find that after removing the forward process, "COUP w/o guidance" (i.e., DiffPure w/o forward process) achieves better robustness, as shown in Fig. 2(b). For theoretical support, we refer to Proposition 4.2, where introducing the forward noise causes the term $\|\epsilon\|$ to double, leading to a larger upper bound of distance (between purified image and input image).

### C.2  SCORE FUNCTION PLAYS A ROLE AS DENOISER

In this section, we will introduce the reason why maximizing the probability density $p(r(\boldsymbol{x}_{\mathrm{adv}}))$ can remove adversarial noise. Then we will discuss the effect of the score function.

Specifically, the training objective (Song et al., 2020b) of score function $\boldsymbol{s}_{\theta}(\boldsymbol{x}_t, t)$ is to mimic $\nabla_{\boldsymbol{x}_t} \log p_{0t}(\boldsymbol{x}_t \mid \boldsymbol{x}_0)$ by score matching (Hyvärinen & Dayan, 2005; Song et al., 2020a)

$$\boldsymbol{\theta}^* = \arg\min_{\boldsymbol{\theta}} \mathbb{E}_t \left\{ \lambda(t) \mathbb{E}_{\boldsymbol{x}_0} \mathbb{E}_{\boldsymbol{x}_t|\boldsymbol{x}_0} \left[ \|\boldsymbol{s}_{\boldsymbol{\theta}}(\boldsymbol{x}_t, t) - \nabla_{\boldsymbol{x}_t} \log p_{0t}(\boldsymbol{x}_t \mid \boldsymbol{x}_0)\|_2^2 \right] \right\}. \tag{26}$$

As for VP-SDE there has $p_{0t}(\boldsymbol{x}_t \mid \boldsymbol{x}_0) \sim \mathcal{N}\left(\sqrt{\alpha_t}\boldsymbol{x}_0, (1-\alpha_t)\boldsymbol{I}\right)$, where $\alpha_t = e^{-\int_0^t \beta(s)\mathrm{d}s}$. Since the gradient of the density given a Gaussian distribution is $\nabla_{\mathbf{x}} \log p(\mathbf{x}) = -\frac{\mathbf{x}-\boldsymbol{\mu}}{\sigma^2}$ where $\boldsymbol{\epsilon} \sim \mathcal{N}(\mathbf{0}, \mathbf{I})$,

thus we have

$$
\begin{aligned}
\boldsymbol{s}_\theta\left(\boldsymbol{x}_t, t\right) &\approx \mathbb{E}_{p_{0t}(\boldsymbol{x}_0)}\left[\nabla_{\boldsymbol{x}_t} \log p_{0t}\left(\boldsymbol{x}_t \mid \boldsymbol{x}_0\right)\right] \\
&= \mathbb{E}_{p_{0t}(\boldsymbol{x}_0)}\left[-\frac{\boldsymbol{x}_t - \sqrt{\alpha_t}\boldsymbol{x}_0}{1 - \alpha_t}\right] \\
&= \mathbb{E}_{p_{0t}(\boldsymbol{x}_0)}\left[-\frac{\sqrt{(1 - \alpha_t)}\boldsymbol{\epsilon}_\theta\left(\boldsymbol{x}_t, t\right)}{1 - \alpha_t}\right] \\
&= \mathbb{E}_{p_{0t}(\boldsymbol{x}_0)}\left[-\frac{\boldsymbol{\epsilon}_\theta\left(\boldsymbol{x}_t, t\right)}{\sqrt{1 - \alpha_t}}\right] \\
&= -\frac{\boldsymbol{\epsilon}_\theta\left(\boldsymbol{x}_t, t\right)}{\sqrt{1 - \alpha_t}},
\end{aligned}
\tag{27}
$$

where $\boldsymbol{\epsilon}_\theta\left(\boldsymbol{x}_t, t\right)$ corresponds to the predicted noise contained in $\boldsymbol{x}_t$, and $\alpha_t$ is the scaling factor. Therefore, the effect of score function $\boldsymbol{s}_{\boldsymbol{\theta}}(\boldsymbol{x}_t, t)$ is to denoise the image $\boldsymbol{x}_t$ meanwhile increase the data likelihood.

### C.3 ADAPTIVE ATTACK DETAILS

In order to evaluate our defense method against strong attacks, we propose an augmented SDE to compute the gradient of COUP for gradient-based attacks. In other words, we expose our purification strategy to the attacker to obtain strict robustness evaluation. In this way, we can make a fair comparison with other adversarial defense methods. In Section 4.3, we discuss the key idea of adaptive attack. Suppose $\hat{\boldsymbol{x}}_{\text{ben}}$ is the input of classifier, $\frac{\partial \mathcal{L}}{\partial \hat{\boldsymbol{x}}_{\text{ben}}}$ can be obtained easily obtained. Then we can get the full gradient $\frac{\partial \mathcal{L}}{\partial \boldsymbol{x}_{\text{adv}}}$ according to the augmented SDE. According to the SDE in Eq. 5 with input $\boldsymbol{x}_{\text{adv}}$ and output $\hat{\boldsymbol{x}}_{\text{ben}}$, the augmented SDE is

$$
\begin{pmatrix} \boldsymbol{x}_{\text{adv}} \\ \frac{\partial \mathcal{L}}{\partial \boldsymbol{x}_{\text{adv}}} \end{pmatrix} = \text{sdeint}\left(\begin{pmatrix} \hat{\boldsymbol{x}}_{\text{ben}} \\ \frac{\partial \mathcal{L}}{\partial \hat{\boldsymbol{x}}_{\text{ben}}} \end{pmatrix}, \tilde{\boldsymbol{f}}, \tilde{\boldsymbol{g}}, \tilde{\boldsymbol{w}}, 0, t^*\right)
$$

where $\frac{\partial \mathcal{L}}{\partial \hat{\boldsymbol{x}}_{\text{ben}}}$ is the gradient of the objective $\mathcal{L}$ w.r.t. the output $\hat{\boldsymbol{x}}_{\text{ben}}$ of the SDE, defined in Eq. 5, and

$$
\begin{aligned}
\tilde{\boldsymbol{f}}([\boldsymbol{x}; \boldsymbol{z}], t) &= \begin{pmatrix} \boldsymbol{f}(\boldsymbol{x}, t) - g(t)^2\{\boldsymbol{s}_\theta\left(\boldsymbol{x}, t\right) + \nabla_{\boldsymbol{x}} \log[f_{\text{cls}}(\boldsymbol{x})]\} \\ \{\frac{\partial \boldsymbol{f}(\boldsymbol{x}, t) - g(t)^2 \boldsymbol{s}_\theta(\boldsymbol{x}, t)}{\partial \boldsymbol{x}} - g(t)^2 \nabla_{\boldsymbol{x}}^2 \log[f_{\text{cls}}(\boldsymbol{x})]\}\boldsymbol{z} \end{pmatrix}, \\
\tilde{\boldsymbol{g}}(t) &= \begin{pmatrix} -g(t)\mathbf{1}_d \\ \mathbf{0}_d \end{pmatrix}, \quad \tilde{\boldsymbol{w}}(t) = \begin{pmatrix} -\boldsymbol{w}(1 - t) \\ -\boldsymbol{w}(1 - t) \end{pmatrix},
\end{aligned}
$$

with $\mathbf{1}_d$ and $\mathbf{0}_d$ representing the $d$-dimensional vectors of all ones and all zeros, respectively. Empirically, we use the stochastic adjoint method (Li et al., 2020) to compute the pathwise gradients of Score SDE.

## D MORE IMPLEMENTATION DETAILS

Our code is available in an anonymous Github repository: `https://anonymous.4open.science/r/COUP-3D8C/README.md`.

### D.1 OFF-THE-SHELF MODELS

We develop our method on the basis of DiffPure: `https://github.com/NVlabs/DiffPure/tree/master`. As for the checkpoint of the off-the-shelf generative model, we use vp/cifar10_ddpmpp_deep_continuous pre-trained from Score SDE: `https://github.com/yang-song/score_sde`. As for the robustness of baselines and the checkpoint of the classifier, it is provided in RobustBench: `https://github.com/RobustBench/robustbench`.

### D.2 KEY CONSIDERATION FOR REPRODUCTION

In Table 1, we experiment with our method on an NVIDIA A100 GPU, while other experiments are tested on an NVIDIA V100 GPU.

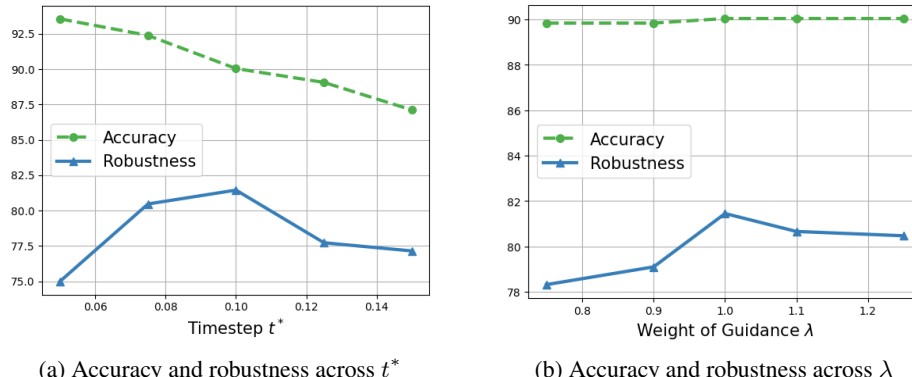

(a) Accuracy and robustness across $t^*$        (b) Accuracy and robustness across $\lambda$

Figure 4: Accuracy and Robustness against APGD-ce under $l_\infty(\epsilon = 8/255)$ threat model for (a) variant purification timestep $t^*$ and (b) variant weight of guidance $\lambda$.

**Random Seed** Consider the setting of DiffPure (Nie et al., 2022), we use seed = 121 and data seed = 0.

**Automatic Mixed Precision (AMP)** In our experiments, we use *torch.cuda.amp.autocast (enabled=False)* to alleviate the loss of precision caused by A100 GPU's automatic use of fp16 for acceleration. The reason is our guidance is small in the integral drift function of each step of SDE. Empirically, we found our method performs even well on V100 GPU, however, it is too computationally expensive. In other words, we may gain better performance than proposed in Table 1 if avoiding accelerating strategies e.g. tested on V100 GPU.

### D.3 Resources and Running Time

To evaluate the robustness of COUP against AutoAttack, we use 2 NVIDIA A100 GPUs and run analysis experiments against APGD-ce attack on 2 NVIDIA V100 GPUs. It takes about 18 days to test our COUP for AutoAttack once (8 days for "rand" mode + 10 days for "standard" mode). For this reason, we do not provide error bars.

## E More Experimental Results

### E.1 Hyperparameters

**Analysis on Purification Timestep** $t^*$ Since the purification timestep $t^*$ is a critical hyperparameter deciding the degree of denoising, we evaluate the robustness against APGD-ce across different $t^*$ and find it performs the best at $t^* = 0.1$. The experimental result in Fig. 4(a) shows that insufficient purification step or "over purification" both leads to lower robustness. This phenomenon strongly supports that balancing denoising and information preservation is very important. Besides, it is intuitive that accuracy decreases as timestep $t^*$ grows.

**Analysis on Guidance Weight** $\lambda$ We experimentally find that COUP obtains the highest robustness against APGD-ce under $\lambda = 1$. That is, the diffusion model and the classifier have equal weight. Note that it implements the same effect as a conditional generative model (according to the Bayes formula: $p(\boldsymbol{x}) \cdot p(y|\boldsymbol{x}) = \frac{p(\boldsymbol{x}|y)}{p(y)}$ since the prior probability $p(y)$ of category $y$ is considered as uniform).

### E.2 Robustness against Salt-and-Pepper Noise Attack

Thanks to the powerful capability of the diffusion model, modeling the data distribution serves as a defensive measure against not only adversarial attacks but also corruptions. Though this is not the main proposal of our main paper, we conduct a preliminary assessment experiment focused on mitigating salt-and-pepper noise.

In detail, we randomly select 500 examples from the CIFAR-10 dataset for evaluation and adopt $t^* = 0.06$ against salt-and-pepper noise attack with $\epsilon = 0.4$. In this context, the robustness of WideResNet-28-10, without purification, stands at 87.80%. Conversely, COUP, without guidance, exhibits a robustness of 92.60%. Remarkably, our implementation of COUP attains a robustness of 93.00%. The outcomes indicate that our COUP is capable of defending against salt-and-pepper noise attacks, demonstrating enhanced performance regarding classifier-confidence guidance.

### E.3 SPEED TEST OF INFERENCE TIME

We do the speed test of the inference time of DiffPure, reverse-time SDE, and COUP on variant $t^*$. The results in Table 2 are tested on an NVIDIA V100 GPU. They show that with larger timestep $t^*$, the inference time will obviously increase. The larger inference time of COUP is caused by computing the gradient of classifier confidence at each diffusion timestep.

Table 2: Inference time (in seconds) different diffusion timestep $t^*$ on one example of CIFAR-10, and the classifier is WideResNet-28-10.

| Method | $t^* = 0$ | $t^* = 0.001$ | $t^* = 0.050$ | $t^* = 0.10$ | $t^* = 0.150$ |
|---|---|---|---|---|---|
| DiffPure | 0.006 | 0.140 | 3.056 | 5.770 | 8.648 |
| COUP w/o Guidance | 0.006 | 0.148 | 3.032 | 6.930 | 8.732 |
| our COUP | 0.006 | 0.170 | 3.894 | 7.662 | 11.134 |

### E.4 PURIFIED IMAGES

We plot the purified images by different purification methods including DiffPure, COUP without Guidance, and our COUP. We follow the setting of 2(b) and use WRN-28-10 as the classifier.

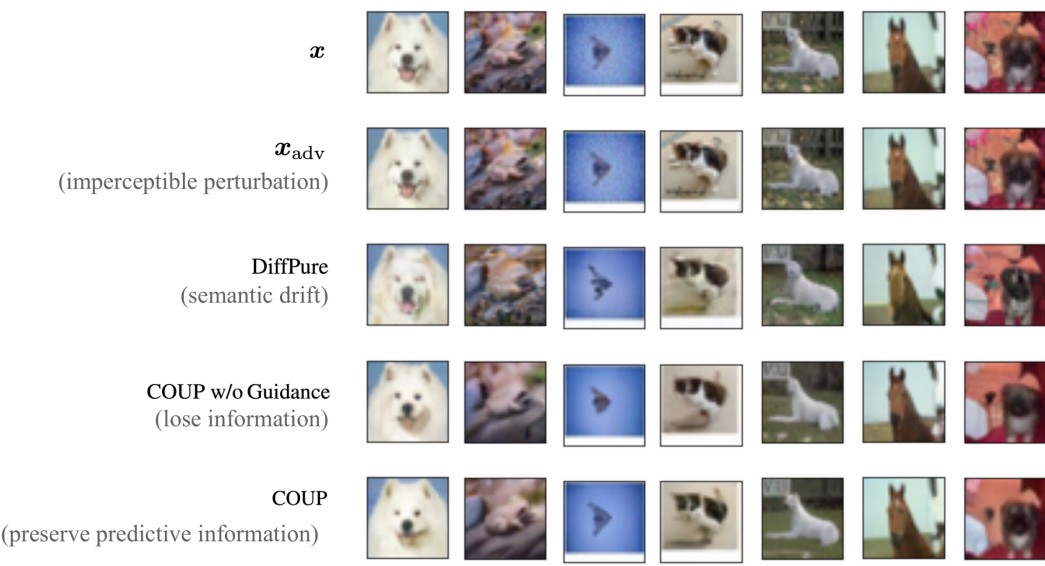

Figure 5: Purified images by DiffPure, SDE, COUP.

As shown in Fig. 5, the purified images of DiffPure suffer from unexpected distortion caused by the forward diffusion process and it of COUP w/o Guidance has lost some details e.g. texture due to its denoising process. Specifically, the 2nd example purified by DiffPure suffers a significant semantic drift from a "frog" to a "bird", and the 3rd example occurs some unexpected feature, which is not beneficial for classification. Moreover, the 3rd image purified by COUP w/o Guidance lost a wing of the airplane and buffed the ears of the cat in the 4th image. However, our COUP has augmented some classification-related features while removing the malicious perturbation.

We further focus on the purification effectiveness of COUP compared with the method without guidance.

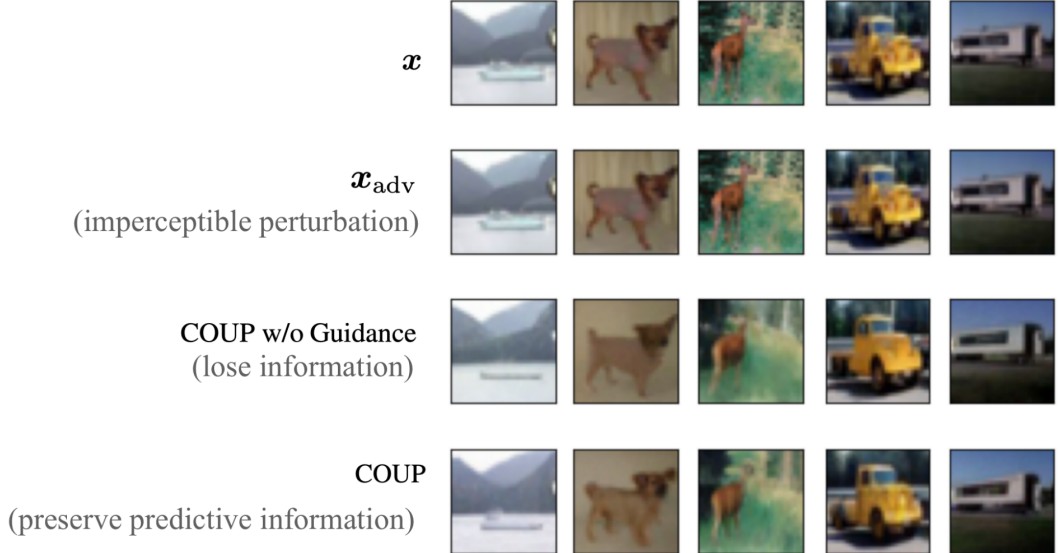

Figure 6: Purified images to show the impact of classifier guidance.

In Fig. 6, we plot the bad cases of COUP w/o Guidance which are correctly classified after adding classifier guidance. It shows that reverse-time SDE does destroy the image details (e.g. the body of the ship in 1st example and the texture of the dog and the details of the track have been destroyed) which is viewed as "over-purification". However, our COUP alleviates this phenomenon (maintaining the predictive information which leads to correct classification).

In conclusion, the results of the example show that

- DiffPure sometimes tends to cause semantic changes due to the diffusion process.
- Using only reverse-time SDE (i.e. COUP w/o Guidance) can avoid semantic drift, however, it faces the problem of information loss in the process of denoising.
- Our method tackles the above problems by removing the forward diffusion process and enhancing the classification-related features.

## F    LIMITATIONS

Despite the effectiveness, our method suffers from high computational costs at inference time. Besides, the hyperparameter timestep $t^*$ and guidance weight $\lambda$ need to be searched empirically, which further increases the calculation overhead.

## G    BROADER IMPACTS

Our method does not create new models, instead, we take advantage of the off-the-shelf generative models and discriminative models. Besides, we design a method for purifying adversarial perturbations, one of the adversarial defenses to improve model robustness. It would not bring risks for misuse.

