# OpenReview forum: "Classifier Guidance Enhances Diffusion-based Adversarial Purification by Preserving Predictive Information"
_ICLR.cc/2024/Conference — ICLR 2024 Conference Withdrawn Submission_

### Official Review · Reviewer_mPZU · 2023-10-31

**Soundness:** 3 good
**Presentation:** 3 good
**Contribution:** 3 good
**Rating:** 6
**Confidence:** 3

**Summary:**

This paper proposes an adversarial purification algorithm called COUP which incorporates the information from the classifier. Intuitively, it can increase the density and at the same time increase the confidence of classifiers. By doing this, the method can achieve better results than previous methods such as DiffPure. Also, the authors provide a theoretical analysis and toy examples to show that the proposed method is well-founded.

**Strengths:**

- The proposed method is simple but effective, classifier guidance is overlooked in DM-based purification
- The paper is well-written and easy to read
- The performance is competitive, better than previous baseline methods

**Weaknesses:**

- Assume we have a stronger diffusion model, which can model the real distribution better, will the classifier-guided purification have smaller privilege, a good choice is the current SOTA EDM [https://github.com/NVlabs/edm]
- Only present results on CIFAR-10, which may not be sufficient.

**Questions:**

- For Section 4.2 line:4 , the input into the reverse SDE should be noised x_adv (pass through forward), or x_adv?
- Can the proposed COUP attack against anti-purification attacks, such as [1, 2] ?
-  What is the bound without classifier guidance, it is better than the boun in proposition 4.2 ?
-  What is the difference between COUP w/o guidance and DiffPure?

[1] Diffusion-Based Adversarial Sample Generation for Improved Stealthiness and Controllability

[2] Diffusion Models for Imperceptible and Transferable Adversarial Attack

---

### Official Review · Reviewer_7jaB · 2023-10-31

**Soundness:** 2 fair
**Presentation:** 3 good
**Contribution:** 2 fair
**Rating:** 5
**Confidence:** 4

**Summary:**

This paper addresses the problem of adversarial purification, finds a drawback of it and then proposes a novel strategy to alleviate the issue.
Adversarial purification methods (Nie et al. 2022) consider a pretrained diffusion model to purify any given adversarial example. Such a mechanism works well if the data distribution is well separated because then the marginal data distribution learned by the diffusion model is equivalent to the conditional distribution of the classifier. However, in many cases, this assumption does not hold and consequently lead to mixing of the samples between neighboring distributions of different classes. This paper proposes a simple mechanism to address this issue—to consider guidance from the classifier during the sampling process.

The authors also utilize a toy example with 1-d Gaussian distribution to present their use-case. Additionally, they perform experiments on widely used classifiers against robust benchmarks under l-2 and l-inf settings.

**Strengths:**

The authors present their ideas in a simple, yet understandable manner. They motivate their use-case using intuitive visualizations and make their case with toy data. They also show how label flip probability is aligned with their method providing a strong motivation for their methodology.

**Weaknesses:**

Although this paper is well written, there are few problems I have here:

Methodology:
1. The method utilizes the ∇x log max y pˆ(y|x) to consider during the purification process. Since x is unknown, r(x_adv) is used to compute this gradient. This tells me that the first projection by the diffusion model is critical for this method to work. The conditional gradient then prevents mixing between different class labels.
2. While this would work well for low adversarial noise, I think this method would struggle under high adversarial noise. With slightly larger noise (l-inf ϵ > 8), the first step of the diffusion model may not project the adversarial example to the right cluster. Consequently, this may lead to mixing between different class labels.
3. Another point is also the amount of overlap between the clusters of different class labels. With a marginal overlap, ∇x log p(x) could be equivalent to ∇x log p(y|x). Only under a strong overlap do they diverge. In ideal settings, not many clusters may have such a strong overlap.

Experiments:
1. If my understanding about the methodology is correct (regarding first projection by diffusion model), then the BPDA attack should be exposed to this hypothesis, instead of the standard setting used. The attacker needs to make sure that the first step is attacked with a higher weight.
2. Table 1 does not show the results for WRN-70-16.

**Questions:**

1. Under the 1-d Gaussian setting, at what point of separation one could see that the gradient of the marginal distribution is equivalent to the gradient of the conditional distribution. This also relates to Figure 1a (visualization), where I believe the overlap is slightly lesser than the 1-d case and it could be that the gradient direction is incorrect for p(x).
2. Another important setting I see missing from the paper is that a simple cooling of the marginal distribution as mentioned in the `Guided Diffusion` paper [1] Appendix G may reduce the overlap of the clusters and prevent the mixing of the samples between different class labels. Why should classifier guidance be better than simple cooling?
3. Table 1 does not show the results for WRN-70-16. It would be intuitive to see the results on more than one classifier.
4. It is not clear to me what are the settings used in Figure 2.

[1] Dhariwal, Prafulla, and Alexander Nichol. "Diffusion models beat gans on image synthesis." Advances in neural information processing systems 34 (2021): 8780-8794.

---

### Official Review · Reviewer_Jfjx · 2023-10-31

**Soundness:** 2 fair
**Presentation:** 3 good
**Contribution:** 3 good
**Rating:** 6
**Confidence:** 3

**Summary:**

This paper considers the issue of adversarial defense, a famous problem that how to improve the robustness of a given model against adversarial attacks. They proposes one kind of adversarial purification method, called COUP algorithm, to approach the goal.

The key ideas and the features in this paper includes:

- COUP uses classifier confidence to guide the adversarial purification process with diffusion models. This helps preserve predictive information while removing adversarial noise.
- It provides theoretical analysis showing classifier guidance can mitigate label shift and avoid misclassification.
- Experiments on CIFAR-10 dataset demonstrate COUP achieves higher adversarial robustness compared to prior purification methods like DiffPure.
- Ablation studies validate the benefits of classifier guidance and removing the forward diffusion process.
Case studies and example purified images provide insights into how COUP works.

**Strengths:**

- Present two theoretical guarantees. Though the first one (Proposition 4.1) works under the case of 1-dim SDE; and the second one (Proposition 4.2) require three bounds $C_s,\,C_p,$ and $C_x$ as conditions.
- Design Case Study and toy experiment (2-Gaussian distribution) to demonstrate that classifier guidance alleviates the information loss, as well as improves adversarial robustness.

**Weaknesses:**

- Lack of experimental results. Maybe consider CIFAR-100 (larger number of classes) as dataset is necessary. Since “classifier guidance” is important in your main subject, the number of classes should be an important variable to be considered. 
- The theoretical results and the presented toy experiment may not fully support the subject, there is a large gap from the practical situation.

**Questions:**

1. In p. 5, the $t$ variable is missing in the RHS of the score function $s_\theta(x,t)=\nabla_x\log\hat{p}(x)$. Does it need to be corrected?
2. In Fig. 2(a), what does the word “disminative” mean? I don’t really understand.
3. How do you evaluate the diffusion-based models like COUP and GDMP? Did you evaluate them under DiffPure’s environment setting?
4. What does the quoted sentence mean: “To focus on the purification process, we do not
consider Brownian motion at inference time.”? Does this mean that you fix a random seed in Case Study?
5. It seems that your proposed Proposition 4.2 looks similar to Theorem 3.2 in DiffPure. Is there any relevance? Also, how to control $C_s,\,C_p,$ and $C_x$?

---

### Official Review · Reviewer_Ac9j · 2023-11-03

**Soundness:** 3 good
**Presentation:** 2 fair
**Contribution:** 2 fair
**Rating:** 3
**Confidence:** 4

**Summary:**

This paper proposes a method to improve the existing solution for adversarial purification. The main idea is to add one more term about the classifier scores to the optimization objective of the diffusion-based model. The goal is to strike a good trade-off between adversarial noise removal and classification information preservation.

**Strengths:**

+ The idea is straightforward and technically sound. It is rationale to add the classifier posterior for better classification accuracy.

+ The paper is clear and easy to follow. The authors clearly presented the motivation for their approach.

+ Experimental results indicate improvements in terms of purification metrics.

**Weaknesses:**

- The proposed method is technically sound but incremental. It is basically doing multi-task learning by simultaneously purifying images and predicting the class labels. Getting improved performance is not surprising at all.

- The motivation is to address the balance between noise removal and information preservation. However, I didn't see too much discussion or design to address this balance. By adding the classification loss, the information preservation can be addressed. But how about the nose removal? Is there any tradeoff between noise removal and information preservation? Any theoretical analysis? Any empirical validation?

- I find the experiments are weak. Very limited baseline methods are compared. More state-of-the-art methods with strong performance should be compared. Only CIFAR-10 is used for experiments. How about other real-world datasets? How about other types of images or data?

- The theory analysis is hard to follow. I tried to read the supplementary but found it needs better organization to make the proof clearer and more concise.

**Questions:**

See weaknesses.